# Effect of Increased Daily Water Intake and Hydration on Health in Japanese Adults

**DOI:** 10.3390/nu12041191

**Published:** 2020-04-23

**Authors:** Yumi Nakamura, Hiroshi Watanabe, Aiko Tanaka, Masato Yasui, Jun Nishihira, Norihito Murayama

**Affiliations:** 1Suntory Global Innovation Center Ltd. Research Institute, 8-1-1 Seikadai, Seika-cho, Soraku-gun, Kyoto 619-0284, Japan; H_Watanabe@suntory.co.jp (H.W.); Norihito_Murayama@suntory.co.jp (N.M.); 2Department of Medical Management and Informatics, Hokkaido Information University, Nishi-Nopporo 59-2, Ebetsu, Hokkaido 069-8585, Japan; aiko.t@do-johodai.ac.jp (A.T.); nishihira@do-johodai.ac.jp (J.N.); 3Department of Pharmacology, Keio University School of Medicine, 35 Shinanomachi, Shinjuku, Tokyo 160-8582, Japan; myasui@a3.keio.jp

**Keywords:** water intake, hydration, blood pressure, body temperature, microbiome

## Abstract

Increased hydration is recommended as healthy habit with several merits. However, supportive data are sparse. To assess the efficacy of increased daily water intake, we tested the effect of water supplementation on biomarkers in blood, urine, and saliva. Twenty-four healthy Japanese men and 31 healthy Japanese women with fasting blood glucose levels ranging from 90–125 mg/dL were included. An open-label, two-arm, randomized controlled trial was conducted for 12 weeks. Two additional 550 mL bottles of water on top of habitual fluid intake were consumed in the intervention group. The subjects drank one bottle of water (550 mL) within 2 h of waking, and one bottle (550 mL) 2 h before bedtime. Subjects increased mean fluid intake from 1.3 L/day to 2.0 L/day, without changes in total energy intake. Total body water rate increased with associated water supplementation. There were no significant changes in fasting blood glucose and arginine vasopressin levels, but systolic blood pressure was significantly decreased in the intervention group. Furthermore, water supplementation increased body temperature, reduced blood urea nitrogen concentration, and suppressed estimated glomerular filtration rate reduction. Additionally, existence of an intestinal microbiome correlated with decreased systolic blood pressure and increased body temperature. Habitual water supplementation after waking up and before bedtime in healthy subjects with slightly elevated fasting blood glucose levels is not effective in lowering these levels. However, it represents a safe and promising intervention with the potential for lowering blood pressure, increasing body temperature, diluting blood waste materials, and protecting kidney function. Thus, increasing daily water intake could provide several health benefits.

## 1. Introduction

Water is fundamental to existence and has numerous roles in the human body. It constitutes 75% of body weight in infants and 50% of body weight in adults, and it is essential for cellular homeostasis and life [1,2]. Water in our body dissolves and transports nutrients and waste products in the blood. When waste products are transported to the kidneys, they are filtered and excreted as urine, maintaining a constant blood concentration. In addition, various biochemical reactions occur in an aqueous solution in the body, thereby maintaining cell activity and regulating body temperature [3]. Indeed, without water, humans can survive only for days. The effects of dehydration are not only acute conditions like heatstroke, but also increased risks of kidney stones, chronic kidney disease, urinary tract infections, cardiovascular diseases, and metabolic disease [4]. Furthermore, a recent report found that mild dehydration affected mood and cognitive functions [5,6,7]. During the past decade, considerable public attention has focused on the importance of adequate hydration. For example, the European Food Safety Authority (EFSA) has revised the existing recommended intake of substances with a physiological effect by including water, since it is essential for life and health. The EFSA has defined adequate intake (AI) as 2.0 L/day for females and 2.5 L/day for males, derived from a combination of observed intake in population groups with desirable osmolarity values of urine and desirable water volumes per energy unit consumed [8]. In addition, the US Institute of Medicine (IOM) defines AI as 3.7 L/day for men and 2.7 L/day for women [9]. However, despite its well-established importance, it is ironically often ignored as a dietary constituent in Japan, because the scientific evidence for the relationship between water intake and health functions is still unclear.

In Europe, a 9-year follow-up study showed that water intake was inversely and independently associated with the risk of developing hyperglycemia [10]. Besides, it is suggested that arginine vasopressin (AVP), released from the posterior pituitary gland, may have a key role in the development of metabolic and cardiovascular disease. One of the many physiological functions of AVP is to maintain constant plasma osmolality in conditions of low water intake by mediating water reabsorption through vasopressin receptor 2 in the renal collecting ducts [11,12]. Furthermore, there are many potential ways in which AVP could influence glucose metabolism; in particular, the vasopressin receptor 1a, which is widely expressed in the body, is involved in glycogenolysis and gluconeogenesis in the liver [13].

Therefore, we focused on the fasting blood glucose level as the primary outcome, but our goal in this study was to comprehend the effect of water intake on human whole-body health. Several interventional studies regarding water intake have been conducted thus far. In a randomized 12-month trial with 1.5 L/day water intake intervention in patients with recurrent urinary tract infections, the mean number of antimicrobial regimens used to treat cystitis episodes was smaller than in the control group [14]. This paper suggested that increased water intake is an effective antimicrobial-sparing strategy to prevent recurrent cystitis in premenopausal women at high risk of recurrence, who drink low volumes of fluid daily. In a randomized trial of a 3-month intervention with 1.5 L/day water intake in patients with headaches, headaches were reduced, and patient quality of life was increased [15]. Furthermore, in a study that increased the amount of water intake over 4 days for young American women who had low levels of fluid intake (1.6 ± 0.5 L/day) found that urine volume increased, urine osmolality decreased, and plasma AVP concentration decreased [16]. In addition, it has been reported that when people with high levels of copeptin, a surrogate marker of AVP, and a routine low level of water intake were given 1.5 L/day of water supplementation for six weeks, both the copeptin level and glucose concentration in plasma decreased [17]. Furthermore, a study on healthy young subjects reported a decrease in 24-h urine osmolality and 24-h urine-specific gravity that was positively and significantly associated with plasma copeptin concentration after continuous water intake of 1, 1.5, or 2 L/day for six weeks [18]. Based on these reports, we can hypothesize that water intake has a good impact on the body’s general health, but studies on the general health effects of increased habitual water intake in healthy adults, or on water supplementation in healthy Asian adults, are yet to be performed. In addition, it has been reported that the gut microbiota changes in order to maintain resistance to the change of fluid balance due to temperature change and exercise [19,20], suggesting the effect of body hydration conditions on the microbiota. However, there are no reports examining the effects of water intake on the intestinal flora.

In this study, we monitored the effects of daily water supplementation on the hydration status; biomarkers in the blood, urine, and saliva; and vital signs of Japanese adults. We also examined the effects on the intestinal flora. This is the first report involving Japanese subjects.

## 2. Materials and Methods

### 2.1. Study Design

A randomized, placebo-controlled, parallel-group study was conducted over a period of 12 weeks. The time period from the initial screening tests to the end of test period was placed between August and December 2018. The study was conducted at Hokkaido Information University, Health Information Science Center (Ebetsu, Hokkaido, Japan).

Interviews were conducted by a research doctor and a nurse during Weeks 0, 4, 8, and 12 to obtain health information and to check physical measurements and vital signs, and perform blood sampling, urinalysis, and salivary sampling. Fecal samples were inspected at Weeks 0 and 12. In addition, the subjects filled in a questionnaire on daily fluid intake (Appendix A) and physical condition starting from one week prior to the study, which is the pre-observation period. The schedule is summarized in Table 1.

### 2.2. Participants

One hundred and seventy-four subjects were recruited to participate in this study. The inclusion criteria for eligible subjects were: Japanese participants with a fasting blood glucose level from 90–126 mg/dL, aged 50–75, and with normal kidney function with an estimated glomerular filtration rate (eGFR) ≧ 60.

The exclusion criteria were:Subjects who are under treatment medication and medical lifestyle advice for diabetes and hypertension;Subjects with serious cerebrovascular, cardiac, hepatic, renal, gastrointestinal, endocrine, and metabolic disorders and/or an infectious disease that requires a report to the authorities;Subjects who experienced an unpleasant feeling during blood drawing;Subjects regularly taking any medicine, functional food, or supplements that affect blood glucose levels—including dietary fiber, such as indigestible dextrin or polyphenol—or any that affect hypertension, such as acetic acid, γ (Gamma)-aminobutyric acid, Tochucha-glycosides, flavonoids, sesamin, chlorella, olive leaf, olive oil, garlic, dietary fiber, tomatoes, folic acid, and hihatsu (long pepper) extract;Heavy smokers (>21 cigarettes/day);Heavy alcohol drinkers (pure alcohol >40 g/day);Subjects with an irregular lifestyle and subjects deemed ineligible by the principal investigator.

Sixty eligible participants were randomly assigned to two groups by the stratified permuted block randomization method. The assignment factors were age, sex, fasting blood glucose level, and systolic blood pressure. The two groups either drank two 550 mL bottles of water per day in addition to their usual fluid intake (intervention group) or drank no additional fluids (control group). After allocation, two participants were withdrawn from intervention group. Fifty-seven subjects participated in the test period.

### 2.3. Intervention

Suntory Tennensui (natural mineral water from the Southern Japanese Alps) was used for water supplementation. Two additional 550 mL bottles of water were given daily on top of habitual fluid intake for 12 weeks in the intervention group. The subjects drank one bottle of water within 2 h of waking, and one bottle 2 h before bedtime.

The method of intake calculation used the following formula:Intake rate (%) = actual number of days of intake/specified number of days of intake × 100
where “actual number of days of intake” constitutes the days on which water supplementation was given, and “specified number of days of intake” was considered the 84 days between Weeks 0 and 12.

### 2.4. Study Outcomes

The primary outcome was the difference in fasting blood glucose levels from baseline at Weeks 4, 8, and 12 between the control and intervention groups. The secondary outcomes were the differences in physical, hematological, and biological assessments.

### 2.5. Physical, Hematological, and Biological Assessments

Blood pressure (BP) and pulse were measured by a nurse with an automatic blood pressure monitor, using the upper arm region of the non-dominant arm. Body composition (body weight, total body water (TBW), extracellular water/total body water (ECW/TBW), body mass index (BMI), and body fat percentage) was measured using InBody770. The TBW rate was calculated as TBW/body weight. We measured axillary temperature upon visitation.

Blood was collected from subjects after a 12-h fast and used for the following hematological examinations: white blood cells, red blood cells, hemoglobin (Hb) and platelet counts, and hematocrit. The biological examinations were as follows: liver function (aspartate aminotransferase, alanine aminotransferase, γ-glutamyltranspeptidase, alanine phosphatase, lactate dehydrogenase, and total bilirubin); renal function (blood urea nitrogen (BUN), creatinine (CRE), and uric acid); lipid profiles (total cholesterol, low-density lipoprotein cholesterol, high-density lipoprotein cholesterol, non-HDL cholesterol, triglycerides, and apolipoproteins B and E); blood glucose profiles (fasting plasma glucose, hemoglobin A1c (HbA1c), glycoalbumin (GA), glucagon, and insulin); and other profiles (serum protein, albumin, cortisol, adrenocorticotropic hormone (ACTH), AVP, creatinine phosphokinase (CPK), serum Na, and C-reactive protein (CRP)).

The eGFR, creatinine clearance, and plasma osmolality were calculated using the following formulas [21].
eGFR (mL/min/*1.73* m^2^) =194 × [CRE]^−1.094^ × [Age]^−0.287^ (for females, ×0.739)
Creatinine clearance (mL/min) = (140 − [Age]) × [body weight]/(72 × [CRE]) (for female, ×0.85)
according to the Cockdroft–Gault equation [22,23].
Plasma osmolality (*mOsm*/kg) = 2 × [serum Na] + [blood glucose]/18 + [BUN]/2.8 

First morning urine samples were collected from subjects. Urine pH, specific gravity, Na, creatinine, albumin, and aquaporin 2 (AQP2) were measured. Urine color was evaluated by eye using a color chart for hydration status evaluation [24,25].

During visits, saliva samples were collected from subjects by having them keep a sponge in their mouth for 1 min. Salivary secretion and IgA were measured.

The hematological tests, urinalysis, and salivary test were performed at Sapporo Clinical Laboratory, Inc. (Hokkaido, Japan).

### 2.6. Record of Fluid and Energy Intake

During the study period, subjects were asked to record the daily intake of test bottled water and fluid in the questionnaire diary for a simplified description of fluid intake status (Appendix A).

In addition, subjects were asked for a detailed description in the descriptive questionnaire three days before each visit. Compliance with water supplementation was checked through the diary. Furthermore, the validity of the diary used in this study was confirmed by correlation with the descriptive questionnaire. Nutritional value was calculated from the descriptive questionnaire, and the energy intake was then calculated.

### 2.7. Microbiome Analysis

Fecal samples were collected from subjects at Weeks 0 and 12. Microbiome analysis was performed using the open-source bioinformatics pipeline, QIIME (Quantitative Insights into Microbial Ecology; Version 1.8.0). In the representative sequence preparation step, CD-HIT-OTU (Version 0.0.1) was used for creating an operational taxonomic unit (OTU). The detailed workflow is as follows. First of all, sequenced paired-end reads were assembled to construct contigs. In the following step, chimeric contigs were removed to the utmost by applying the CD-HIT-OTU algorithm. After that, the remaining contigs were clustered into OTUs with 97% sequence similarity. To acquire taxonomic information for each OTU, representative sequences were aligned to the Greengenes 16S rRNA database (g_13.8) by PyNAST (Version 1.2.2) (GitHub, Inc., San Francisco CA, USA) and assigned to its database for classification by RDP classifier (Version 2.2) (Michigan State University Board of Trustees, East Lansing, MI, USA ). Likewise, a homology search was conducted for representative sequences and assigned to DDBJ 16S ribosomal RNA database by BLASTN (Version 2.2.20).

Comparative statistical analysis was conducted for the relative abundance of bacteria in each sample.

### 2.8. Ethics

This study was conducted according to the guidelines laid down in the Declaration of Helsinki (revised by the Fortaleza General Meeting of the World Medical Association), and all procedures involving human subjects were approved by the ethics committee of Hokkaido Information University (Ebetsu, Hokkaido, Japan; Approved on 19 July 2018; approval number: 2018-09). Written informed consent was obtained from all subjects. This study is also in compliance with the Ethical Guidelines for Medical and Health Research Involving Human Subjects (2014 Ministry of Education, Culture, Sports, Science and Technology/Ministry of Health, Labour and Welfare Ministerial notification No. 3). This trial was registered at the University Hospital Medical Information Network-Clinical Trials Registry (UMIN-CTR) (www.umin.ac.jp/ctr/index.htm; registered on 4 August 2018; registration number: UMIN000033587).

### 2.9. Statistical Analysis

Statistical analysis was performed using JMP ver.14.0.0 (SAS Institute Inc., Cary, NC, USA). Numerical data were expressed as mean ± standard deviation (SD). Statistical comparisons were performed using two-way analysis of variance between the groups using variation from Week 0 of each index, Tukey’s test for comparison between groups, or a paired *t*-test adjusted by the Bonferroni method for subgroup variation. For microbiome analysis, we used the paired *t*-test for subgroup variation and the two-sample *t*-test for comparison between groups. *P* < 0.05 was considered statistically significant.

## 3. Results

### 3.1. Flow and Baseline Characteristics

Out of the 174 people recruited, we selected 60 people to participate in the study. However, as two people dropped out of the study after allocation, we conducted the study with 58 subjects in total, 30 subjects in control group and 28 subjects in intervention group. All subjects completed the study. After completion of the study, three subjects were removed from the analysis for influencing the data based on the following reasons: the long-term use of antihypertensive agents and analgesics (*n* = 2), or asthma symptoms being seen during the study period (*n* = 1). Finally, statistical analysis was done on 27 subjects in the control group and 28 subjects in the intervention group. Figure 1 shows the flow diagram of the study. No adverse events due to the intervention were observed in this study.

Table 2 shows the baseline characteristics of the study subjects. The average number of each parameter at the screening is shown for each group. The average age of participants, in terms of mean ± SD, was 63.9 ± 6.1 (control group) and 63.4 ± 6.9 (intervention group). There were no significant differences between the two groups in terms of age, gender, fasting blood glucose, and systolic blood pressure as assignment factors. Females were the majority subjects in this study. On the other hand, the control group tended to have significantly higher weight and BMI than the intervention group (Table 2).

### 3.2. Change in Daily Water Intake and Body Hydration

The intake rate in the intervention group was 99.7% overall. We checked the amount of daily fluid intake by the questionnaire diary (Appendix A). The mean fluid intake during the pre-observation period and test period (Week 0 to Week 12) in the control group was almost unchanged, ranging from 1428–1445 mL/day, whereas in the intervention group, there was an increase of 656 mL from 1342–1998 mL/day due to the intervention. The difference in daily fluid intake between the control group and intervention group increased significantly in the test period (mean ± SD: 1444 ± 566 mL vs. 1998 ± 488 mL, respectively, *p* < 0.001) (Figure 2a). Nevertheless, there was almost no difference in the pre-observation period (1428 ± 540 mL vs. 1342 ± 611 mL, respectively, *p* = 0.96). With respect to the fluid intake by the intervention group, daily water intake increased from 490–1244 mL, and that of other beverages remained almost unchanged (Figure 2b).

We examined the validity of the fluid intake in the questionnaire diary in this study by linear regression, providing y = 0.96x + 5.88 (x = fluid intake in questionnaire diary as the explanatory variable, y = fluid intake in the descriptive questionnaire as the objective variable for the general linear model, R = 0.93, *p* < 0.001).

With respect to body hydration, the TBW rate increased significantly in the intervention group at Weeks 8 and 12 (Figure 3a). There were no significant changes in the ECW/TBW (Figure 3b).

### 3.3. Effects of Increased Water Intake

There were no significant changes in the fasting blood glucose level, which is the primary outcome (Figure 4a). AVP, which was considered as one of the mechanisms of lowering blood glucose by water intake, showed no significant change (Figure 4b). However, among other blood glucose profiles, HbA1c and GA decreased significantly in the intervention group (Appendix A).

The systolic blood pressure (SBP) in the intervention group decreased over time from Week 0 to Week 12, from 123.8 ± 20.5 to 117.4 ± 15.7 mmHg (mean ± SD, *p* = 0.01) (Figure 5a). There were no significant changes in the diastolic blood pressure (DBP) (Figure 5b). In terms of body temperature, the control group showed a 0.2 °C decrease at Week 8, whereas the intervention group showed a rise of 0.8 °C (*p* = 0.004) (Figure 5c).

In terms of renal function, in the intervention group, BUN decreased from 15.6 ± 4.4 to 13.8 ± 4.4 mg/dL from Week 0 to Week 8 (*p* < 0.001) (Figure 6a). In the control group, eGFR decreased significantly from 66.6 ± 7.9 to 64.5 ± 7.2 mL/min/1.73 m^2^ from Week 0 through Weeks 4 and 8 to Week 12 (*p* = 0.003). In the intervention group, on the other hand, there was no significant change in eGFR (Figure 6b).

While the calculated plasma osmotic pressure increased significantly in the control group from 294.6 ± 2.9 to 297.2 ± 2.8 mOsm/kg H_2_O through the test period, no such changes were seen in the intervention group (Figure 6c). In other words, in the intervention group, we saw a suppression of the rise in plasma osmotic pressure (*p* = 0.004).

The urine AQP2 concentration corrected by urine CRE (AQP2/CRE), which is an indicator of the re-absorption of water in the kidneys, showed a significant decrease in the intervention group from 0.06 ± 0.06 to 0.04 ± 0.05 ng/mL/CRE from Weeks 0 to 8, compared to the control group (*p* = 0.001) (Figure 6d). The other parameters regarding liver function, the lipid profiles, the hematological examinations, the other blood tests such as cortisol, ACTH, CKP, and CRP, the urinalysis, and the salivary tests are summarized in Appendix A. None of the parameters fluctuated outside of the standard values, and there were no safety issues.

There were also no changes in energy intake that were due to water Appendix A.

### 3.4. Change of Microbiome

There were five out of the 108 types of intestinal bacterium at the genus level that showed significant differences within and between groups due to water supplementation, and 17 out of 564 species identified by the homology search of OTU representative sequences in the DDBJ 16S database showed the same behavior (Appendix A). Of these, a slight correlation was seen in the intervention group between changes in blood pressure and changes in the Psudoflavonifractor capillosus bacterial count (R = 0.42) (Figure 7). Kineothrix sp. (R = 0.36), Feacalibacterium prausnitzii (R = 0.38), and Ruminococcaceae (R = 0.34) showed weak correlations between changes in body temperature and changes in bacterial count (Figure 8). Even though we were unable to identify the genus of Ruminococcaceae, we believe it is something other than Anaerotruncus, Butyricicoccus, Faecalibacterium, Oscillospira, Ruminococcus, gnavas, bromii, or ID1.

## 4. Discussion

An open-label, two-arm, randomized controlled trial was conducted for 12 weeks. Two additional 550 mL bottles of water on top of habitual fluid intake were consumed in the intervention group. The subjects drank one bottle of water (550 mL) within 2 h of waking, and one bottle (550 mL) 2 h before bedtime. Subjects increased mean fluid intake from 1.3 L/day to 2.0 L/day.

The major effects of water supplementation in this study were a decrease in blood pressure and an increase in body temperature. Putative mechanisms by which blood pressure decreased include: (a) the removal of excess sodium and water due to improved kidney function; (b) changes in the secretion of hormones involved in blood pressure increase, such as renin and aldosterone; and (c) a decrease in peripheral vascular resistance due to waste excretion. A conceivable possibility was that waste products were excreted or blood components were diluted, thereby at least reducing vascular resistance, since there was a reduction in BUN and hematocrit (Appendix A). However, in order to further examine the mechanism, it is necessary to examine the secretion of hormones related to blood pressure, urine volume, the excretion of urine components, and blood pressure. We also investigated the intestinal flora as another possible mechanism. In the intervention group, there was a weak correlation between a decrease in blood pressure and rise in *Psudoflavonifractor capillosus.* Although the function of this bacterium remains unclear, it may be involved in the mechanism of blood pressure reduction. We do not think that the result was affected by the high blood glucose in subjects, as we were unable to see a correlation between baseline blood glucose and fluctuation in blood pressure.

With respect to the rise in body temperature, as the baseline temperature differs between groups, this may be a regression to the mean. The present study was conducted between September and December in Sapporo, and the mean monthly atmospheric temperature varied from 18.9 to −1.0 °C. Whereas the control group showed a decrease in body temperature with the change of season, the intervention group seems to have had a suppression of this decrease. Impaired thermoregulation due to dehydration [26], the slower reactivity of skin blood vessels with age [27], and decreased blood flow in the skin [28] lead to a loss of heat from the body. We think that by inducing a well hydrated state in elderly people, we were able to improve the decline in body temperature control and suppress the heat loss. As hyperthermia is known to affect metabolism and immune function, being able to maintain or raise the body temperature by water supplementation may enhance metabolism and improve immune function. In addition, we examined how the mechanisms of body temperature elevation may be related to the change in the intestinal flora. As a result, we saw a weak correlation between the rise in body temperature from water intake and decreases in *Kineothrix sp.*, *Feacalibacterium prausnitzii*, and *Ruminococcaceae* levels (Figure 8). These bacteria may have been involved in the mechanism of temperature regulation.

Another suggested effect is the suppression of kidney function decline. As the BUN level decreased, we think that waste material may have been diluted or excreted. Furthermore, while the eGFR significantly decreased in the control group from Week 0 to Week 8, there was no such significant change in the intervention group. In other words, we think that water supplementation helped to suppress the decline in kidney function that occurs due to dehydration during the winter [29]. Although kidney function is known to decline with age, habitual water intake may suppress the decrease in kidney function.

Even though the water supplementation in this study was 1.1 L/day in total, this was actually an addition of 656 mL/day on top of pre-intervention intake, so the actual intervention level was 60%. Since the intervention group originally took 490 mL of water during the pre-observation period, the actual increase was considered to be 656 mL despite the intervention using 1.1 L in a day. In previous studies on water intake interventions, a 1.5 L/day intervention was actually 842 mL/day [15], and other studies had 1.5 L/day intervention where the actual water intake was 1.0 L/day [30]. Therefore, the actual intervention level was approximately 56%–67%, which is similar to in the present case. Furthermore, as water supplementation did not cause changes in the consumption of other drinks and energy intake, we were able to investigate its influence precisely. The present subjects originally had an intake of 1.3 L/day, which we believe was less than the daily intake level recommended by EFSA and IOM. The total amount due to water supplementation was 2.0 L/day, which was closer to the AI recommended by EFSA. In terms of the safety of the intake method, as mentioned by the EFSA, water intoxication with life-threatening hypo-osmolarity is rare, with rapid rehydration occurring after near-drowning in fresh water or the overconsumption of water that exceeds the kidney’s maximal excretion rate of 0.7–1.0 L/hour [8]. In the present intervention, the excretion rate was 0.275L/hour and was within this criterion. There were no adverse events due to the increased intake level, and there were no safety issues associated with water supplementation, as the fluctuation of each parameter was within the reference range.

With respect to body hydration due to water supplementation, the TBW rate increased, but there was no increase in ECW/TBW. We think that extra body water from water supplementation was largely retained in the cells. The retention of intra-cellular water leads to the maintenance of the homeostasis of various reactions in the cell. Furthermore, as BUN, hematocrit, and urine specific gravity, which are used as indicators of dehydration, are reduced, we believe this indicates a state of improved hydration. In addition, while the calculated plasma osmolality increased significantly in the control group through the test period, the increase was suppressed in the intervention group. In particular, the elderly are known to have a higher steady-state plasma osmotic pressure than younger people [9,31], and they tend to become dehydrated easily during the summer and winter, further raising the plasma osmotic pressure and affecting sympathetic nerve stimulation and hormone secretion [31,32,33]. By suppressing the rise in plasma osmotic pressure, it may be possible to normalize this impact. In other words, good hydration could lead to the maintenance of homeostasis in various physiological functions such as nutrient transport, the removal of waste, and hormone secretion.

In addition, there was a significant decrease in urine AQP2/CRE in the intervention group between Weeks 0 and 8. AQP2 is translocated to the apical membrane in the renal collecting duct, triggered upon detection by vasopressin receptor 2. AVP is secreted from the pituitary gland with an increase in plasma osmotic pressure, and water is reabsorbed via AQP2 to regulate the body fluid volume [34,35]. Urine AQP2 is known to correlate with the amount of AQP2 translocated to the membrane and is used as an indicator of the reabsorption of water [36,37]. The decrease in urine AQP2/CRE between Week 0 and Week 8 can be explained by the reduction of water reabsorption due to good hydration. On the other hand, BUN and urine AQP2/CRE rose in the intervention group at Week 12. BUN is thought to have been strongly influenced by factors other than water intake such as seasonality and/or lifestyle changes. AQP2/CRE changes might have happened as compensation or to change the setpoint to maintain homeostasis. Further studies are needed to address this significant issue.

The fasting blood glucose level, which is the primary outcome in this study, did not show significant changes due to the intervention. We focused on AVP as a mechanism of blood glucose reduction through water intake. Higher AVP levels, as a result of low water intake, stimulate vasopressin 1a and 1b receptors and may result in higher plasma glucose levels [38], which may be reversed by the water-induced reduction of AVP. However, the initial AVP level in the intervention group was 2.2 ± 2.3 pg/mL, which was low for the subject at baseline. It is surmised that as this subject did not have a fluctuating range of AVP, there was no impact on the blood glucose level either. In an ongoing study in Europe, which involves subjects with low water intake and high concentrations of copeptin, an alternative marker for AVP, hydration (a 1.5 L increase in daily water intake) vs. control therapy (no change in water intake) will be tested, with the difference in fasting plasma glucose as the primary outcome (NCT03422848). In this manner, when the study is limited to people with high AVP, intervention by water intake may decrease the blood glucose level, but this study involved people just with high blood glucose levels, and there was no blood glucose-reducing effect. On the other hand, other blood glucose profiles, such as HbA1c and GA, decreased significantly in the intervention group. Since HbA1c provides a more accurate overall picture of what our average blood sugar levels have been over a period of weeks/months, these data could be reflective of improvements to glucose metabolism due to the effects of continued daily water supplementation. With regard to the influence on glucose metabolism with a conditional effect, it is necessary to conduct further studies under separate conditions, such as in the study in Europe (NCT03422848).

Regarding the baseline characteristics in this study, weight and BMI were significantly different between the groups (Table 2). They could affect some of parameters and might mask intervention effects, which is the limitation of this study. However, we confirmed that the major effects—which are on systolic blood pressure, body temperature, renal function, and microbiota—were not confounded by weight, according to multiple regression analysis.

In the present study, as a result of 12 weeks of water supplementation (two additional 550 mL bottles of water per day) when waking and before sleeping, the ECW/TBW remained unchanged, but the TBW rate increased. As a whole-body effect, we confirmed that subject health improved, based on indicator fluctuations, such as the decrease in blood pressure, rise in body temperature, dilution of waste materials, and suppression of kidney function decline. At present, it is not clear whether these effects are due to the increase in water intake levels, or because of the timing of the study. However, we have shown that even healthy adults can gain several health benefits from changing their habitual water intake. This is the first paper, especially in Asia, to have investigated general health parameters in healthy people. Furthermore, it is the first paper that mentioned a decrease in blood pressure and rise in body temperature from water supplementation. As there were limitations regarding the details of the mechanisms and the sample size, we anticipate further results from large-scale studies like the European study (NCT03422848).5. Conclusions

Habitual water supplementation after waking up and before bedtime in healthy subjects with slightly elevated fasting blood glucose levels is not effective in lowering these levels. However, it represents a safe and promising intervention with the potential for lowering blood pressure, increasing body temperature, diluting blood waste materials, and protecting kidney function. Thus, increasing daily water intake could provide several health benefits.

## Figures and Tables

**Figure 1 nutrients-12-01191-f001:**
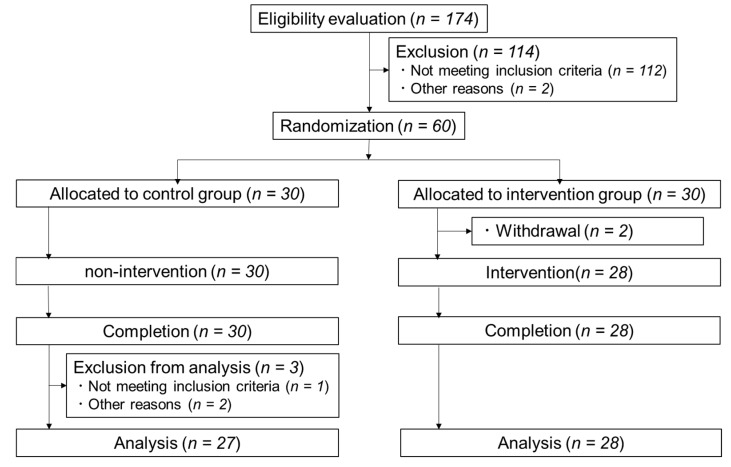
Flow diagram for the involvement of subjects during the trial.

**Figure 2 nutrients-12-01191-f002:**
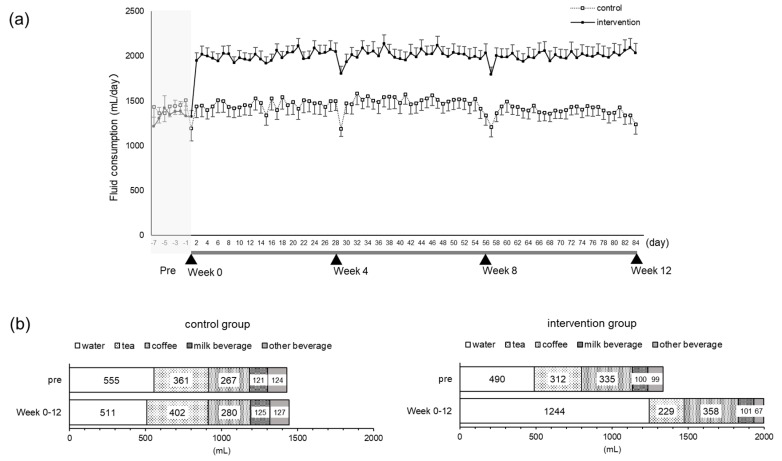
Daily fluid intake. Total daily fluid intake from the pre-observation period to the end of the test period. The dotted line indicates the control group. The solid line shows the intervention group. Mean ± SEM (**a**). The mean amount of each kind of fluid intake during the pre-observation period and the test period (Week 0 to Week 12) (**b**).

**Figure 3 nutrients-12-01191-f003:**
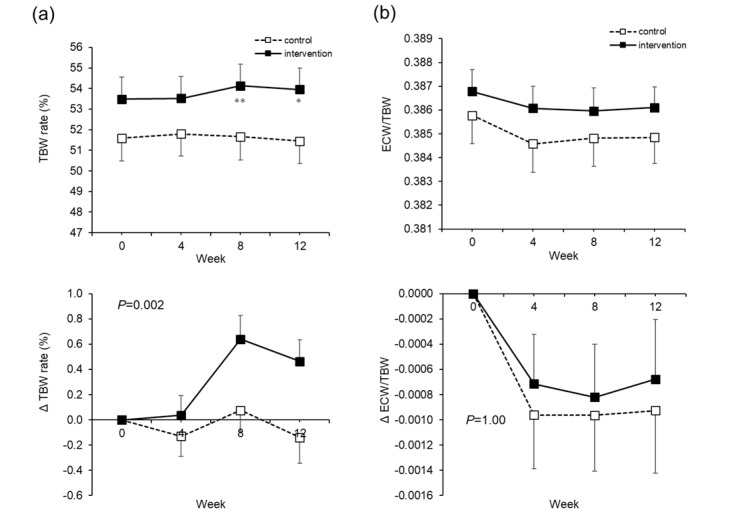
Total body water (TBW) rate and extra-cellular water (ECW)/TBW. The actual level (upper) and changes (lower) in the TBW rates (**a**). The actual level (upper) and changes (lower) in ECW/TBW (**b**). The dotted line is the control group. The solid line is the intervention group. Mean ± SEM. The *P* value is the difference between the groups for the change from Week 0 to Week 12. Two-way analysis of variance was performed. Paired *t*-test adjusted by the Bonferroni method was performed for intragroup comparisons from Week 0. * *p* < 0.05, ** *p* < 0.001.

**Figure 4 nutrients-12-01191-f004:**
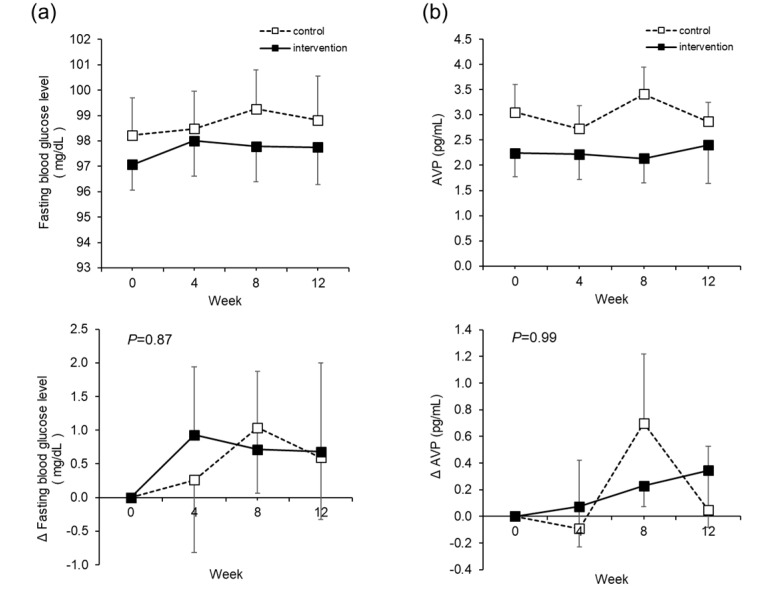
Fasting blood glucose level and arginine vasopressin (AVP). The actual level (upper) and changes (lower) in the fasting blood glucose level (**a**). The actual level (upper) and changes (lower) in AVP (**b**). The dotted line is the control group. The solid line is the intervention group. Mean ± SEM. The *p* value is the difference between the groups for the change from Week 0 to Week 12. Two-way analysis of variance was performed. Paired *t*-test adjusted by the Bonferroni method was performed for intragroup comparisons from Week 0. There was no significant difference.

**Figure 5 nutrients-12-01191-f005:**
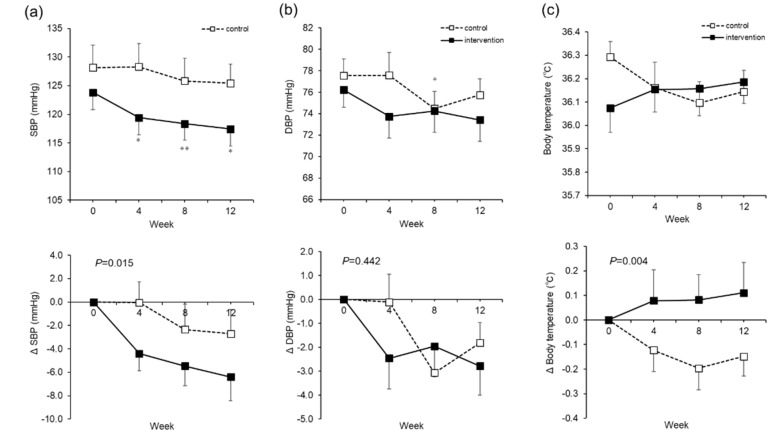
Systolic blood pressure (SBP), diastolic blood pressure (DBP), and body temperature. The actual level (upper) and changes (lower) in SBP (**a**). The actual level (upper) and changes (lower) in DBP (**b**). Body temperature. The actual level (upper) and changes (lower) in body temperature. The dotted line is the control group (**c**). The dotted line is the control group. The solid line is the intervention group. Mean ± SEM. The *P* value is the difference between the groups for the change from Week 0 to Week 12. Two-way analysis of variance was performed. * Paired *t*-test adjusted by the Bonferroni method was performed for intragroup comparisons from Week 0. * *p* < 0.05, ** *p* < 0.001.

**Figure 6 nutrients-12-01191-f006:**
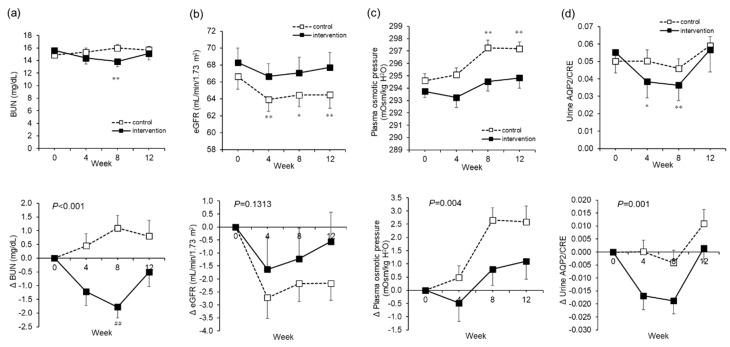
Renal functions, plasma osmotic pressure, and urine aquaporin2/creatinine (AQP2/CRE). The actual level (upper) and changes (lower) in blood urea nitrogen (BUN) (**a**). The actual level (upper) and changes (lower) in the estimated glomerular filtration rate (eGFR) (**b**). The actual level (upper) and changes (lower) in plasma osmotic pressure (**c**). The actual level (upper) and changes (lower) in urine AQP2/CRE (**d**). The dotted line is the control group. The solid line is the intervention group. Mean ± SEM. The *P* value is the difference between the groups for the change from Week 0 to Week 12. Two-way analysis of variance was performed. * Paired *t*-test adjusted by the Bonferroni method was performed for intragroup comparisons from Week 0. * *p* < 0.05, ** *p* < 0.001, ## *p* < 0.001.

**Figure 7 nutrients-12-01191-f007:**
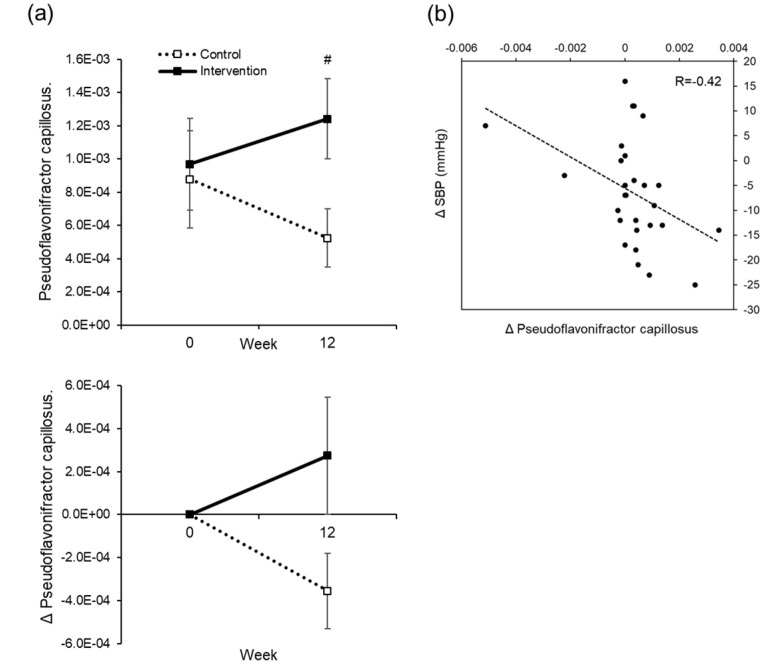
Correlation between the change in systolic blood pressure (SBP) and changes in intestinal bacteria. The actual level (upper) and changes (lower) in the OTU of Psudoflavoniflactor capillosus. The dotted line is the control group. The solid line is the intervention group. Mean±SEM (**a**). # *t*-test of two independent samples, for comparison between groups at each time point, # *p* < 0.05. A correlation diagram between the changes in OTU of Psudoflavoniflactor capillosus, and changes in SBP between Weeks 0 and 12, in the intervention group (**b**). R = Correlation coefficient.

**Figure 8 nutrients-12-01191-f008:**
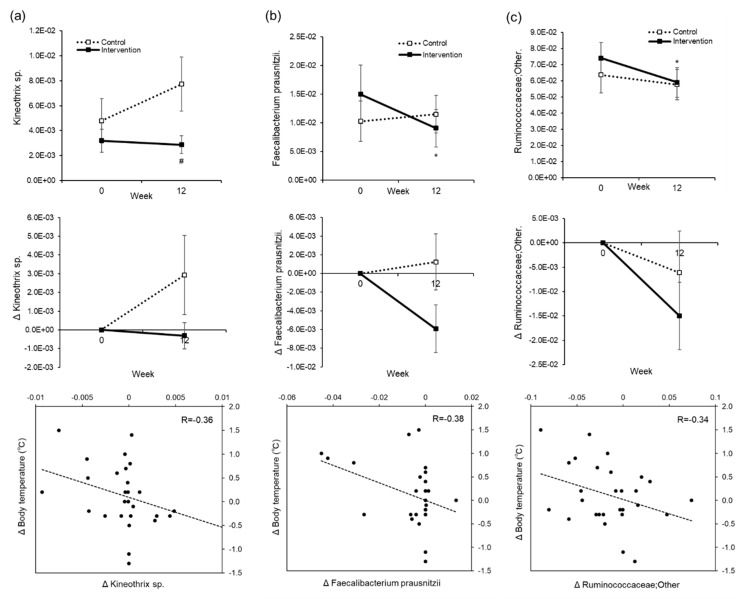
Correlation between changes in body temperature and changes in intestinal bacteria. The actual level (upper), changes (middle), and correlation between changes (lower) in the operational taxonomic unit (OTU) of Kineothrix xp. (**a**), Feacalibacterium prausnizzii (**b**), and Ruminococcaceae (**c**). The dotted line is the control group. The solid line is intervention group. Mean ± SEM. * Paired *t*-test was performed for intragroup comparisons from Week 0. * *p* < 0.05. # *t*-test of two independent samples, for comparison between groups at each time point, # *p* < 0.05.

**Table 1 nutrients-12-01191-t001:** Clinical trial schedule.

Parameters	Guidance Agreement Screening	Pre-Observation Period (1 Week)	Test Period				
			Start Date of Intervention	Intervention Period			
			Week 0	Week 2	Week 4	Week 8	Week 12
**Visit**	●	□	●	●	●	●	●
**Informed consent**	●	□	□	□	□	□	□
**Medical interview**	●	□	●	□	●	●	●
**Physical measurement**	●	□	●	□	●	●	●
**Vital signs**	●	□	●	□	●	●	●
**Blood sampling**	●	□	●	□	●	●	●
**Urinalysis**	●	□	●	□	●	●	●
**Salivary sampling**	□	□	●	□	●	●	●
**Fecal sampling**	□	□	●	□	□	□	●
**Diary record**	□	●	●	●	●	●	●

**Table 2 nutrients-12-01191-t002:** Baseline characteristics of the study subject.

Parameters	Control (*n* = 27)	Intervention (*n* = 28)	*p* Value
Age (years)	63.9 ± 6.1	63.4 ± 6.9	0.708
Gender (male/female)	13/14	11/17	0.508
Height (cm)	163.2 ± 10.3	161.0 ± 8.9	0.409
Weight (kg)	62.9 ± 12.5	56.5 ± 10.8	0.048
BMI (kg m^−2^)	23.4 ± 2.8	21.7 ± 3.0	0.027
Body fat (%)	27.3 ± 6.8	25.1 ± 6.3	0.223
Fasting blood glucose (mg/dL)	96.5 ± 5.5	95.3 ± 3.5	0.354
Systolic blood pressure (mmHg)	132.6 ± 18.8	127.3 ± 18.7	0.302

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
