# Peer review of "Effect of Increased Daily Water Intake and Hydration on Health in Japanese Adults"

_nutrients, 2020, doi:10.3390/nu12041191_

Round 1

Reviewer 1 Report

This is a very interesting, well-written paper which highlights the crucial role that increased water consumption plays in several health indices, (including kidney function, blood pressure and others), highlighting in parallel the need for more research concerning the importance of this "forgotten" nutrient. Moreover, i believe that the present study makes a significant contribution in this specific scientific area.

Despite the amount of time and effort put forth by the authors which is evident throughout the manuscript, there are few points that should be assessed by the authors.

I would suggest that the manuscript should be sent back to authors for some changes:

General/specific Comments

  • It is not clear throughout the text, and especially in the abstract, how exactly the increase in water consumption was achieved. I would suggest to make the intervention more understandable for the reader like you did in lines 128-130.
  • There is a confusion with the exact number of the participants in the study. Furthermore, it is not clear until you reach in the Results section the sex of the participants. Please clarify that in the entire manuscript.
  • Please "re-construct" Table 2. It is suggested to present all the numbers with only one decimal point. What's the meaning of presenting age for example like 63.89 or 63.36, or height as 163.15 etc. Moreover, it is suggested to present males/females as exacts numbers and not as %. It doesn't makes sense that men participants for example were 39.29% out of 28 in total. Consequently, it should be mentioned in the manuscript that the majority of the subjects was females.
  • It is not reasonable for the reviewer, why the authors chose to focus in fasting blood glucose levels, while at the same time glycated haemoglobin (HbA1c) was measured. It is known that HbA1c provides a more accurate overall picture of what average blood sugar levels have been over a period of weeks/months, thus this could strengthen significantly their results, especially when a significant decrease in the intervention group was observed (line 260).
  • It is suggested to refer in the results sections to other measured parameters as well, like creatinine, SGOT/AST, γ-GT etc.
  • It is suggested to change the structure of the Discussion section. Emphasizing to the main outcomes at its beginning, rather than its limitations (commonly at the end), is a more common tactic which also makes the entire section more understandable for the reader.
  • Lines 382, 392. Reference of specific companies, brands etc, despite being credible, should be avoided.
  • Although not mandatory, it is suggested to include a Conclusion paragraph to the manuscript.

Author Response

Point 1:

As your suggestion, we described more in detail about how the water intervention, as below: ‘Two additional 550 mL bottles of water on top of habitual fluid intake were consumed in intervention group. The subjects drank one bottle of water (550 mL) within 2 hours after waking, and one bottle (550 mL) 2 hours before bedtime. (lines 18-20, 340-342)

Point 2:

We are sorry about it. We added explanation about the number of the participants in entire manuscript.

Also, we mention it in abstract as below: ‘24 healthy Japanese men and 31 healthy Japanese women’. (lines 16-17)

Point 3:

・We changed the number to one decimal point. (line 226 and in Table2)

・In order to explain what Table 2 suggests, we added the description as below: ‘The average number of each parameter at the screening was shown for each group.’ (lines 225-226)

・We changed the present males/females to exact number in Table 2.

・We mentioned it as below: ‘Females were the majority subject in this study.’ (line 228)

Point 4:

The reason why we focused in fasting blood glucose levels was it was a primary outcome in this study. Indeed, HbA1c was significantly decreased in the intervention group. As you suggested, we changed and mentioned it in Discussion part, as below: ‘Since HbA1c provides a more accurate overall picture of what our average blood sugar levels have been over a period of weeks/months, these data could be reflected improvement of glucose metabolism due to the effect of continuing daily water supplementation.’ (lines 438-440)

Point 5:

As your suggestion, we mentioned other parameters as below: ‘The other parameters about liver function, lipid profiles, hematological examinations, blood others such as cortisol, ACTH, CKP, and CRP etc., urinalysis, and salivary test were summarized in Table S1.’ (lines 307-309)

Point 6:

Thank you for your suggestion. We reconstructed Discussion section. (lines 340-497)

Point 7:

As your suggestion, we changed it as ’the study in Europe (NCT03422848)’ (lines 430, 443, and 496)

Point 8:

As your suggestion, we added Conclusion paragraph after Discussion. (lines 498-503)

Reviewer 2 Report

An interesting and well designed study. Informative to understand the health impacts of water supplementation with specific health markers. Systolic blood pressure and body temperature changes are an interesting find and in the counter direction to what I would anticipate. 

Minor points to consider:

  • Please describe how you randomized your sample, what was your method.
  • Were the data collectors or researchers blind to participants allocated groups? Please outline justification why they were not. 
  • Your baseline characteristics (Table 2) highlight significant difference between your groups in weight and BMI, is this not a confound? Please justify why you did not counterbalance anthropometric measures between groups at recruitment. Please outline how you accounted for these baseline differences in your statistical analyses? 

Author Response

Point 1:

We randomized by stratified permuted block randomization method. We described it in Materials and Methods. (lines 123-124)

Point 2:

Yes, data collectors and researchers were blind until data fixed. After allocation in the blind, only the independent person who manage the sending of test bottled water got information about intervention group. The participants knew which one during test period because they should know they do normal life (control) or water supplementation (intervention).

Point 3:

Because we couldn’t set over 5 factors as the assignment factors, we chose age, sex, fasting blood glucose level and systolic blood pressure preferentially. As you mention, anthropometric measures between groups were significantly different as the results. To confirm the confound of weight, we performed multiple regression analysis using each amount of change. As the results, some of parameters such as physical measurement including TBW rate and ECW/TBW, TC, LDL, Non-HDL-C, ApoB, Albumin, creatinine clearance, AST, ALT, g-GTP, total bilirubin, and salivary s-IgA were affected by weight, but the others were not. We confirmed the major effects in this study, which are systolic blood pressure, body temperature, renal function, and microbiota, were significantly explained by intervention, but not weight. Therefore, we consider that the difference of weight baseline indeed affected on some of parameters and might mask intervention effects, while the major effects in this study were not confound of weight. We described it in Discussion. (lines 444-448)

Round 2

Reviewer 1 Report

The authors have responded efficiently to all of my comments